# Large Language Models as Analogical Reasoners

**Michihiro Yasunaga,**[1,2]  **Xinyun Chen,**[1]  **Yujia Li,**[1]  **Panupong Pasupat,**[1]  **Jure Leskovec,**[2]
**Percy Liang,**[2]  **Ed H. Chi,**[1]  **Denny Zhou**[1]

[1] Google DeepMind  [2] Stanford University

myasu@cs.stanford.edu, {xinyunchen,dennyzhou}@google.com

## Abstract

Chain-of-thought (CoT) prompting for language models demonstrates impressive performance across reasoning tasks, but typically needs labeled exemplars of the reasoning process. In this work, we introduce a new prompting approach, **analogical prompting**, designed to automatically guide the reasoning process of large language models. Inspired by analogical reasoning, a cognitive process in which humans draw from relevant past experiences to tackle new problems, our approach prompts language models to self-generate relevant exemplars or knowledge in the context, before proceeding to solve the given problem. This method presents several advantages: it obviates the need for labeling or retrieving exemplars, offering generality and convenience; it can also tailor the generated exemplars and knowledge to each problem, offering adaptability. Experimental results show that our approach outperforms 0-shot CoT and manual few-shot CoT in a variety of reasoning tasks, including math problem solving in GSM8K and MATH, code generation in Codeforces, and other reasoning tasks in BIG-Bench.

## 1 Introduction

Large language models (LLMs) demonstrate strong performance across various tasks (Brown et al., 2020; Chowdhery et al., 2022; Liang et al., 2022; Qin et al., 2023). Recently, chain-of-thought (CoT) prompting has demonstrated LLMs' abilities to tackle complex tasks, such as solving math problems, by prompting them to generate intermediate reasoning steps (Wei et al., 2022b; Kojima et al., 2022). For instance, common methods like few-shot CoT (Wei et al. 2022b; Figure 1, middle) make LLMs generate reasoning steps by offering a few exemplars of question-rationale-answer triplets; 0-shot CoT (Kojima et al. 2022; Figure 1, left) aims for the same objective by offering instructions like "think step by step." These studies highlight the importance of devising effective methods to guide LLMs to reason.

However, the existing CoT paradigm faces two key challenges: providing *relevant* guidance or exemplars of reasoning, and minimizing the need for manual *labeling*. Specifically, 0-shot CoT offers generic reasoning guidance, which may not suffice for complex tasks like code generation (§6). Few-shot CoT provides more detailed guidance but demands labeled exemplars of the reasoning process, which can be costly to obtain for every task. This raises a research question: can we achieve the best of both worlds and automate the generation of relevant exemplars to guide LLMs' reasoning process?

In this work, we propose **analogical prompting**, a new prompting approach that automatically guides the reasoning process of LLMs. Our inspiration comes from analogical reasoning in psychology, a concept where humans draw from relevant past experiences to tackle new problems (Vosniadou & Ortony, 1989). For instance, when given a new math problem (e.g., finding the area of a square given four points in a coordinate system; Figure 1), humans often think about "Do I know a related problem?" (Polya, 2004) and recall how they solved related problems in the past (e.g., finding the area of a square with a known side length) to derive insights for solving the new problem. They also recall high-level knowledge, such as the need to find the side length to calculate a square's area. Our idea is to prompt LLM generation to mimic this reasoning process to effectively solve new problems.

Concretely, given a problem to solve, we prompt LLMs to self-generate relevant exemplars in the context, using instructions like "# Recall relevant problems and solutions:...", and then proceed to solve the original problem (Figure 1, 2). We can also prompt the LLM to generate high-level knowledge that complements specific exemplars, using instructions like "# Provide a tutorial:..." (Figure 3). This turns out to be particularly useful for complex tasks like code generation (see §6). Notably, our method can operate in a single prompt, generating knowledge, exemplars, and a solution to the initial

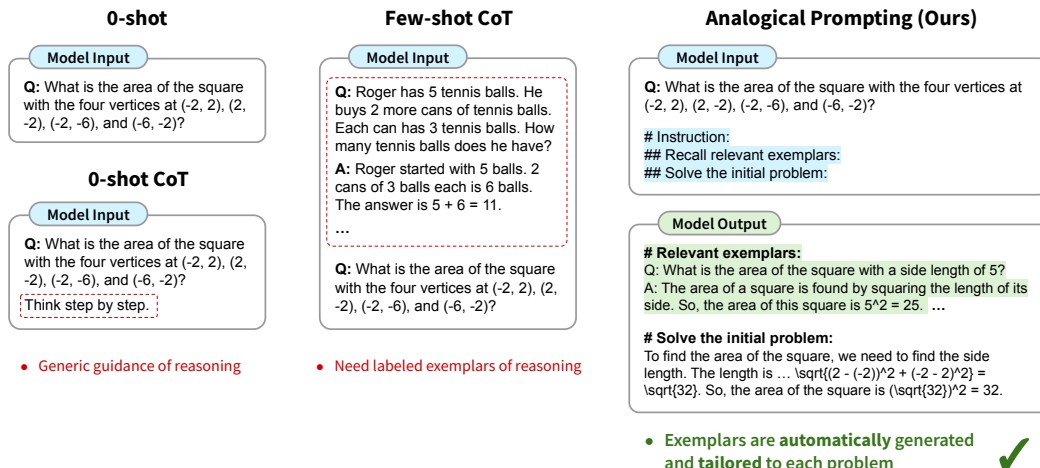

Figure 1: **Overview of our approach, analogical prompting**. *Left*: Existing methods for prompting LLM to reason are either generic (0-shot CoT) or demand labeled exemplars (few-shot CoT). *Right*: Given a problem, our method prompts LLMs to **self-generate** relevant exemplars before solving the problem. This eliminates the need for labeling and also tailors the exemplars to each individual problem. See Figure 3 for a sample prompt where the LLM self-generates both knowledge and exemplars.

problem end-to-end in one pass. The underlying idea here is that modern LLMs have already acquired knowledge of various problems during training. Explicitly prompting them to recall relevant problems and solutions in the context guides LLMs to perform in-context learning to solve new problems.

Our proposed approach offers several advantages. It self-generates exemplars and obviates the need for manually labeling reasoning exemplars for each task, addressing the challenges faced by 0-shot and few-shot CoT. Furthermore, the self-generated exemplars are tailored to individual problems, such as 'geometry' or 'probability', rather than generic 'math problems'. This can simplify the complexity associated with recent CoT techniques that retrieve relevant exemplars from external data (Zhang et al., 2022b; Shum et al., 2023).

We evaluate the proposed approach in various reasoning-intensive tasks, including mathematical problem solving in GSM8K (Cobbe et al., 2021) and MATH (Hendrycks et al., 2021b), code generation in Codeforces, and other reasoning tasks in BIG-Bench (Srivastava et al., 2022). We use several base LLMs: GPT-3.5, GPT-4 (OpenAI, 2023; Ouyang et al., 2022), and PaLM2 (Anil et al., 2023). Experimental results show that our approach outperforms 0-shot CoT and few-shot CoT across a range of tasks and base LLMs, achieving an average accuracy gain of +4%. Notably, our approach improves performance on datasets involving diverse types of reasoning, such as MATH (including algebra, probability, geometry, etc.) and Codeforces (involving dynamic programming, graph algorithms, etc.). This result suggests the effectiveness of generating tailored exemplars for individual problems to guide the reasoning process of LLMs.

## 2 RELATED WORKS

### 2.1 LARGE LANGUAGE MODELS AND PROMPTING

A language model estimates probabilities over text. Recent research has scaled up these models from hundreds of millions (Devlin et al., 2019) to hundreds of billions of parameters (Brown et al., 2020) and expanded training data to include web texts and instruction data (Gao et al., 2020; Ouyang et al., 2022; Chung et al., 2022). These advances have made large language models proficient in various NLP tasks.

LLMs with hundreds of billions of parameters demonstrate in-context learning and few-shot learning abilities (Brown et al., 2020; Liu et al., 2022; Su et al., 2022; Mishra et al., 2022; Wei et al., 2022a; Yasunaga et al., 2023; Shi et al., 2023). They use input prompts (instructions or a few exemplars) to guide LLMs to generate desired responses for tasks, marking the advent of the prompting era. Our approach harnesses the in-context learning abilities of LLMs to guide their reasoning process using self-generated exemplars.

Closely related to ours are works that perform self-generation in LLM prompting (Sun et al., 2022; He et al., 2023; Kim et al., 2022; Li et al., 2022a). For instance, Sun et al. (2022) prompts LLMs to recite relevant facts in context for open-domain question answering. Our idea of self-generating exemplars is related to recitation, but focuses on recalling problem-solving and reasoning processes rather than factual knowledge.

## 2.2 CHAIN-OF-THOUGHT PROMPTING

Chain-of-thought (CoT; Wei et al. 2022b) is a prompting strategy that guides LLMs to produce intermediate reasoning steps towards a final answer, enhancing problem-solving performance. Common instances of CoT include 0-shot CoT (Kojima et al., 2022) and few-shot CoT Wei et al. (2022b).

**0-shot CoT** prompts LLMs with a general instruction like "think step by step" to produce intermediate reasoning steps. **Few-shot CoT** achieves stronger performance by providing multiple exemplars of reasoning process (question–rationale–answer), leveraging LLMs' in-context learning abilities. However, it requires labeled exemplars. Our approach tackles this challenge by prompting LLMs to self-generate exemplars.

Within few-shot CoT, the original approach employs a fixed set of labeled exemplars for all test problems. Recent work explores **retrieval-based CoT**, which aims to obtain more relevant exemplars from external data for each problem (Zhang et al., 2022b; Shum et al., 2023). While our work shares the goal of providing relevant exemplars, instead of retrieval, we make LLMs *self-generate* exemplars. Self-generation offers several advantages: it is simpler, as it does not require external data retrieval, and it is more versatile, as it can produce not only specific exemplars but also broader insights or knowledge that complement them. Empirically, our generation-based CoT outperforms retrieval-based CoT, especially with larger base LLMs, while retrieval-based CoT excels with smaller base LLMs (§6).

Finally, there are other techniques for enhancing CoT, such as self-consistency (Wang et al., 2022) and least-to-most (Zhou et al., 2022). Our work can complement and integrate with these efforts (Table 8). Please see §A for additional related works.

## 3 PRELIMINARIES

We focus on problem-solving tasks, where the objective is to produce a solution $y$ for a given problem statement $x$, such as mathematical questions or code generation specifications. The solution may include both the intermediate reasoning steps or rationale $r$ and the final answer $a$.

A prompting method $\phi$ is a function that maps a problem statement $x$ into a specific textual input $\phi(x)$ for an LLM, which then generates a solution $\hat{y} = \text{LLM}(\phi(x))$. For instance,

- In 0-shot prompting, $\phi$ directly yields $x$.
- In 0-shot CoT, $\phi$ supplements $x$ with a general instruction, such as "[$x$] think step by step".
- In few-shot CoT, $\phi$ supplements $x$ with several labeled exemplars, $\{(x_i, r_i, a_i)\}_{i=1}^{K}$, such as "[$x_1$] [$r_1$][$a_1$]...[$x_K$][$r_K$][$a_K$] [$x$]".

Our aim is to design a prompting method $\phi$ that enhances the accuracy of solutions LLMs generate.

## 4 APPROACH

We introduce **analogical prompting**, a new prompting approach that automatically provides exemplars to guide LLMs' reasoning process. Inspired by how humans recall relevant past experiences when tackling new problems, our approach makes LLMs self-generate relevant exemplars or knowledge in context, before proceeding to solve the problem (Figure 1, right). We present two techniques to achieve this: self-generated exemplars (§4.1) and self-generated knowledge + exemplars (§4.2).

### 4.1 SELF-GENERATED EXEMPLARS

Our approach is based on the idea that modern LLMs possess a broad range of problem-solving knowledge acquired during training. Explicitly prompting them to recall or generate relevant problems and solutions in context aids LLMs to perform in-context learning to solve new problems.

Specifically, given a target problem to solve $x$, our prompt augments it with instructions like:

```
# Problem: [x]
# Relevant problems: Recall three relevant and distinct problems. For each
problem, describe it and explain the solution.
# Solve the initial problem:
```

For a concrete example, see Figure 2. The LLM first generates several ($K$) relevant exemplars in the form of question-rationale-answer sequences ("`# Relevant problems:`" part of the instruction). Then the model proceeds to solve the initial problem, leveraging the recalled exemplars in the context ("`# Solve the initial problem:`" part of the instruction). Note that all these instructions are provided within a single prompt, allowing the LLM to generate relevant problems and solution to the initial problem in one continuous pass. Using '#' symbols in the prompt (e.g., '`# Relevant Problems`') helps LLMs structure the response better.

Below are key technical decisions we made:

- Generating relevant and *diverse* exemplars is important: To achieve this, we explicitly include an instruction in the prompt, such as "generate problems that are distinct from each other" (e.g., Figure 2). This step is crucial as some LLMs have a tendency to repetitively generate identical problems, which can be misleading when solving the target problem.
- Single-pass vs. independent exemplar generation: An alternative approach is to independently generate exemplars by separately sampling them from the LLM and then re-prompting the LLM with all the exemplars. While this method does work, our current single-pass prompt approach achieves comparable performance and offers greater convenience, eliminating the need for multiple prompts. Consequently, we have chosen to adopt the single-pass method.
- The number of exemplars to generate ($K$): Through experimentation, we have found that generating $K = 3$ to 5 exemplars works the best (more details in §6.5).

Our approach offers two advantages. It offers detailed exemplars of reasoning without manual labeling, addressing the challenges in 0-shot and few-shot CoT. The generated exemplars are tailored to individual problems (e.g., 'geometry' or 'probability'), offering more relevant guidance than traditional few-shot CoT, which uses fixed exemplars (e.g., general math problems; Figure 1, middle).

## 4.2 Self-generated knowledge + exemplars

While generating exemplars is useful, in complex tasks like code generation, LLMs may overly rely on the low-level exemplars and fail to generalize when solving the target problems. To address this challenge, we also allow LLMs to self-generate high-level takeaways that complements the exemplars, which we refer to as "knowledge." Specifically, we enhance the prompt with an additional instruction like the following. For a concrete example, see Figure 3.

```
# Tutorial: Identify core concepts in the problem and provide a tutorial.
```

One technical consideration is whether to generate knowledge before or after exemplars. We found that generating knowledge before exemplars yields superior results (Table 7). By generating knowledge first, LLMs identify the core concepts of the problem. This, in turn, helps LLMs generate exemplars that align more closely in terms of the fundamental problem-solving approaches rather than surface-level lexical similarities. For further discussion, please refer to §6.2.

## 5 Experimental Setup

### 5.1 Tasks

We evaluate the proposed approach in diverse reasoning-intensive tasks, including mathematical problem solving, code generation, and other reasoning tasks like logical and temporal reasoning.

**Mathematical problem solving.** We use popular benchmarks, GSM8K (Cobbe et al., 2021), comprising elementary math word problems, and MATH (Hendrycks et al., 2021b), consisting of advanced math problems from high school math competitions. For each problem, we obtain an output from LLMs using a temperature of 0, and report the accuracy.

**Code generation.** Code generation involves synthesizing programs to solve algorithmic problems. Competitive programming is especially challenging, requiring reasoning about various algorithms like dynamic programming and graphs (Li et al., 2022b; Kulal et al., 2019; Yasunaga & Liang, 2020).

As a benchmark, we collected competitive programming problems from codeforces.com (details in §B). We focus on level-A problems published in 2023 to prevent test set contamination (Magar & Schwartz, 2022). Each problem comprises a problem statement, which serves as input to LLMs, and a set of test cases to assess generated code. The correctness of code is determined by whether it passes all test cases.

```
┌─ Our Prompt: Self-generate Exemplars ──────────────────────────────────────────┐
│                                                                                 │
│ Your task is to tackle mathematical problems. When presented with a math problem, recall relevant problems as examples. Afterward, │
│ proceed to solve the initial problem.                                           │
│                                                                                 │
│ # Problem:                                                                      │
│ An airline serves a dinner to all the passengers on an airplane. They get their choice of steak or fish. Three steak meals and three │
│ fish meals are set aside for the six-member crew. If the meals are distributed to the crew members randomly, what is the probability │
│ that both pilots get the fish?                                                  │
│                                                                                 │
│ # Instructions:                                                                 │
│ ## Relevant Problems:                                                           │
│ Recall three examples of math problems that are relevant to the initial problem. Your problems should be distinct from each other and │
│ from the initial problem (e.g., involving different numbers and names). For each problem: │
│ - After "Q: ", describe the problem                                             │
│ - After "A: ", explain the solution and enclose the ultimate answer in \boxed{}. │
│                                                                                 │
│ ## Solve the Initial Problem:                                                   │
│ Q: Copy and paste the initial problem here.                                     │
│ A: Explain the solution and enclose the ultimate answer in \boxed{} here.       │
└─────────────────────────────────────────────────────────────────────────────────┘
```

```
┌─ Output by GPT3.5-turbo ─────────────────────────────────────────────────────────┐
│ ┌ ─ ─ ─ ─ ─ ─ ─ ─ ─ ─ ─ ─ ─ ─ ─ ─ ─ ─ ─ ─ ─ ─ ─ ─ ─ Self-generated Exemplars ─ ─ ┐ │
│ │ Relevant Problems:                                                            │ │
│ │ Q: In a bag, there are 5 red balls and 3 blue balls. If two balls are drawn at random without replacement, what is the probability │ │
│ │ that both balls are red?                                                      │ │
│ │ A: The total number of balls is 8. The probability of drawing a red ball on the first draw is 5/8. After one red ball is drawn, there │ │
│ │ are 4 red balls left out of 7 total balls. Therefore, the probability of drawing a second red ball is 4/7. The probability of both │ │
│ │ events happening is the product of the individual probabilities, so the probability of drawing two red balls is (5/8) * (4/7) = │ │
│ │ \boxed{\frac{5}{14}}.                                                         │ │
│ │                                                                               │ │
│ │ Q: A box contains 10 red marbles and 5 blue marbles. If three marbles are drawn at random without replacement, what is the probability │ │
│ │ that all three marbles are red?                                               │ │
│ │ A: The total number of marbles is 15. The probability of drawing a red marble on the first draw is 10/15. After one red marble is │ │
│ │ drawn, there are 9 red marbles left out of 14 total marbles. Therefore, the probability of drawing a second red marble is 9/14. After │ │
│ │ two red marbles are drawn, there are 8 red marbles left out of 13 total marbles. Therefore, the probability of drawing a third red │ │
│ │ marble is 8/13. The probability of all three events happening is the product of the individual probabilities, so the probability of │ │
│ │ drawing three red marbles is (10/15) * (9/14) * (8/13) = \boxed{\frac{24}{91}}. │ │
│ │                                                                               │ │
│ │ Q: A bag contains 4 green marbles, 3 red marbles, and 2 blue marbles. If two marbles are drawn at random without replacement, what is │ │
│ │ the probability that both marbles are green?                                   │ │
│ │ A: The total number of marbles is 9. The probability of drawing a green marble on the first draw is 4/9. After one green marble is │ │
│ │ drawn, there are 3 green marbles left out of 8 total marbles. Therefore, the probability of drawing a second green marble is 3/8. The │ │
│ │ probability of both events happening is the product of the individual probabilities, so the probability of drawing two green marbles │ │
│ │ is (4/9) * (3/8) = \boxed{\frac{1}{6}}.                                       │ │
│ └ ─ ─ ─ ─ ─ ─ ─ ─ ─ ─ ─ ─ ─ ─ ─ ─ ─ ─ ─ ─ ─ ─ ─ ─ ─ ─ ─ ─ ─ ─ ─ ─ ─ ─ ─ ─ ─ ─ ─ ┘ │
│                                                                                 │
│ Let's solve the following math problem:                                         │
│ An airline serves a dinner to all the passengers on an airplane. They get their choice of steak or fish. Three steak meals and three │
│ fish meals are set aside for the six-member crew. If the meals are distributed to the crew members randomly, what is the probability │
│ that both pilots get the fish?                                                  │
│ A: There are a total of 6 crew members, so the probability of the first pilot getting the fish is 3/6. After one fish meal is given to │
│ the first pilot, there are 2 fish meals left out of 5 total meals. Therefore, the probability of the second pilot getting the fish is │
│ 2/5. The probability of both events happening is the product of the individual probabilities, so the probability that both pilots get │
│ the fish is (3/6) * (2/5) = \boxed{\frac{1}{5}}.                               │
└─────────────────────────────────────────────────────────────────────────────────┘
```

Figure 2: Actual example of our prompt (top) and LLM output (bottom) for MATH task. ***Top***: Our prompt supplements the problem statement with instructions to **generate relevant exemplars** and then solve the problem. ***Bottom***: Exemplars generated by GPT3.5-turbo are indeed relevant to the problem, focusing on probability. It then accurately solves the problem. See §D.1 for the complete prompt and output. Using '#' symbols in the prompt (e.g., '# Relevant Problems') helps LLMs structure the response better.

In line with existing work on code generation (Li et al., 2022b; Chen et al., 2023), we report the Acc@1 and Acc@10 metrics. Acc@$k$ measures whether at least one of the $k$ sampled model outputs is correct. For each problem, we sample 10 outputs from LLMs, using a temperature of 0.7.

**Other reasoning tasks.**    We further evaluate on various reasoning tasks in BIG-Bench (Srivastava et al., 2022; Suzgun et al., 2022): word sorting, logical deduction five objects, temporal sequences, reasoning about colored objects, and formal fallacies. These tasks are diverse and may not have dedicated training data, so they align well with our approach of self-generating custom exemplars. For each problem, we obtain an output from LLMs using a temperature of 0, and report the accuracy.

## 5.2    MODELS

We experiment with several base LLMs: GPT-3.5-turbo, GPT-4 (OpenAI, 2023; Ouyang et al., 2022) (version 0613, accessed in June–September 2023), and PaLM 2-L (Anil et al., 2023).

## 5.3    METHODS TO COMPARE

We compare the following prompting methods, including ours.

**0-shot and 0-shot CoT.**    These methods, like ours, do not use labeled exemplars. We aim to show that our method offers more tailored guidance for LLM reasoning and yields superior task performance.

---

**Our Prompt: Self-generate Exemplars + Knowledge**

```
Your goal is to write Python3 code to solve competitive programming problems. Given a problem, explain the core concepts in it and provide
other relevant problems. Then solve the original problem.

# Problem:
You are given a sequence a_1, a_2, …, a_n. Each element of a is 1 or 2. Find out if an integer k exists so that the following conditions are
met. 1 ≤ k ≤ n-1, and a_1 · a_2 … a_k = a_{k+1} … a_n. If there exist multiple k that satisfy the given condition, print the smallest.

# Instruction:
## Algorithms:
Identify the core concepts or algorithms used to solve the problem.

## Tutorial:
Write a tutorial about these algorithms.

## Example Problems:
Provide three examples of relevant competitive programming problems that involve these algorithms. For each problem, describe the problem,
explain the solution in detail, and then write the correct Python3 code.

## Python3 code to solve the original problem:
- Explanation of the solution:
- Python3 code to solve the problem:
```

---

**Output by GPT3.5-turbo**

*Self-generated Knowledge*

```
## Algorithms: The core algorithm for solving this problem is the prefix product algorithm.

## Tutorial: The prefix product algorithm is used to calculate the product of all the elements in a prefix of an array. The algorithm works
by initializing a variable "prefix" to 1, and then iterating through the array from left to right. At each iteration, the current element is
multiplied with the prefix, and the result is stored in the prefix variable. This way, the prefix variable keeps track of the product of all
the elements in the prefix of the array. The time complexity of this algorithm is O(n), where n is the size of the input array. ......
```

*Self-generated Exemplars*

```
## Example Problems:
1. Problem: Given an array of integers, find the product of all the elements in each subarray of size k.
   Solution: We can use the prefix product algorithm to solve this problem. We initialize a prefix variable to 1, and then iterate through
the array from left to right. At each iteration, we multiply the current element with the prefix, ......

def product_of_subarrays(arr, k):
    prefix = 1
    result = []
    for i in range(len(arr)):
        prefix = prefix * arr[i]
        if i >= k-1:
            result.append(prefix)
            prefix = prefix // arr[i-k+1]
    return result

2. Problem: Given an array of integers, find the longest subarray with a product less than or equal to a given target.
   Solution: ......

3. Problem: Given an array of integers, find the smallest k such that the product of all the elements in the prefix of the array is equal to
the product of all the elements in the suffix of the array.
   Solution: ......
```

```
## Python3 code to solve the original problem:
Explanation of the solution:
  - Initialize prefix and suffix variables to 1, and k to -1.
  - Calculate the product of all the elements in the sequence using the prefix product algorithm.
  - Iterate through the sequence from right to left.
    - Update the prefix and suffix variables.
    - If the prefix is equal to the suffix, update k to the current index.
  - Print the value of k.

t = int(input())
for _ in range(t):
    n = int(input())
    arr = list(map(int, input().split()))
    prefix, suffix = 1, 1
    k = -1
    for i in range(n):
        prefix = prefix * arr[i]
    for i in range(n-1, 0, -1):
        prefix = prefix // arr[i]
        suffix = suffix * arr[i]
        if prefix == suffix:
            k = i
    print(k)
```

Figure 3: Actual example of our prompt (top) and LLM output (bottom) for the Codeforces task. **Top**: Our prompt supplements the problem statement with instructions to **generate knowledge** (e.g., tutorials on core concepts) **and relevant exemplars**, followed by solving the original problem. **Bottom**: The knowledge and exemplars generated by GPT3.5-turbo are indeed relevant to the problem to solve, focusing on the prefix product algorithm. The final code generated by the LLM effectively applies the algorithm to solve the problem. See §D.3 for the complete prompt and output.

**Few-shot CoT.** This is the standard few-shot CoT, using a fixed set of reasoning exemplars across all test problems within a dataset. For the GSM8K and MATH datasets, as their training sets include solutions labeled with reasoning steps, we use $K = 5$ exemplars from these training sets. For the other datasets, we use $K = 3$ manually-annotated exemplars. We aim to show that our method, which *self*-generates exemplars, can match or surpass this baseline, which uses *labeled* exemplars.

**Few-shot retrieved CoT.** Instead of using a fixed set of exemplars, for each test problem, we dynamically retrieve relevant labeled problem-solution pairs from the train set for each test problem. Specifically, we

| Prompting Method | GSM8K Accuracy | | | MATH Accuracy | |
|---|---|---|---|---|---|
| | GPT3.5-turbo | text-davinci-003 | PaLM2 | GPT3.5-turbo | PaLM2 |
| 0-shot | 75.0% | 14.8% | 60.8% | 33.0% | 27.1% |
| 0-shot CoT | 75.8% | 50.3% | 78.2% | 33.9% | 29.8% |
| 5-shot CoT | 76.7% | 54.0% | 80.7% | 34.9% | 34.3% |
| **Ours: Self-generated Exemplars** | **77.8%** | **61.0%**[†] | **81.7%** | **37.3%** | **34.8%** |

Table 1: **Performance on mathematical tasks, GSM8K and MATH**. Our prompting method, which self-generates exemplars, outperforms baselines such as 0-shot CoT and few-shot CoT. [†]For text-davinci models, we use an in-context demonstration of how to generate exemplars. For the other models, we do not.

| Prompting Method | GPT3.5-turbo-16k | | GPT4 | |
|---|---|---|---|---|
| | Acc@1 | Acc@10 | Acc@1 | Acc@10 |
| 0-shot | 8% | 24% | 16% | 30% |
| 0-shot CoT | 9% | 27% | 16% | 29% |
| 3-shot CoT | 11% | 27% | 17% | 31% |
| **Ours: Self-generated Exemplars** | 13% | 25% | 17% | 32% |
| **Ours: Self-generated Knowledge + Exemplars** | **15%** | **29%** | **19%** | **37%** |

Table 2: **Performance on Codeforces code generation task**. Our prompting method outperforms baselines such as 0-shot CoT and few-shot CoT. Moreover, self-generating knowledge provides additional gains over self-generating exemplars, demonstrating its usefulness for the challenging Codeforces task.

| Prompting Method | Word sorting | Logical deduction five objects | Temporal sequences | Reasoning about colored objects | Formal fallacies |
|---|---|---|---|---|---|
| 0-shot | 66.8% | 30.0% | 40.4% | 50.4% | 53.6% |
| 0-shot CoT | 67.6% | 35.2% | 44.8% | 61.6% | 55.6% |
| 3-shot CoT | 68.4% | 36.4% | **58.0%** | 62.0% | 55.6% |
| **Ours: Self-generated Exemplars** | **75.2%** | **41.6%** | 57.6% | **68.0%** | **58.8%** |

Table 3: **Performance on BIG-Bench reasoning tasks** in accuracy. GPT3.5-turbo is used as the base LLM. Across diverse tasks, our method outperforms baselines (0-shot CoT) and is competitive with manual 3-shot CoT.

use Sentence-BERT (Reimers & Gurevych, 2019) to encode each problem statement. For each problem in the test set, we retrieve the top $K = 5$ similar problems from the training set based on cosine similarity.

**Our method.** We let LLMs self-generate $K = 5$ exemplars for GSM8K and $K = 3$ exemplars for MATH and BIG-Bench tasks. For Codeforces, we self-generate both knowledge and $K = 3$ exemplars.

## 6 RESULTS

### 6.1 MAIN RESULTS

**Mathematical problem solving.** Table 1 presents results for GSM8K and MATH tasks. Our prompting method, which self-generates exemplars, outperforms baselines such as 0-shot CoT and few-shot CoT. The improvement over few-shot CoT is notable for the MATH task, which involves a range of reasoning types, including algebra, probability, and geometry. This aligns with our approach of crafting tailored exemplars for each problem.

Figure 1 and 2 provide qualitative examples of GPT3.5-turbo outputs generated using our prompt. In both examples, the LLM indeed generates relevant exemplars (geometry problems in Figure 1. probability problems in Figure 2), and subsequently produces correct solutions. In contrast, in the standard few-shot CoT (Figure 1, middle), the exemplars are math-related (e.g., algebra) but may not always match the test problem (e.g., geometry), as the dataset contains diverse test problems.

**Code generation.** Table 2 presents results for Codeforces task. Our prompting method outperforms baselines such as 0-shot CoT and few-shot CoT in both GPT3.5-turbo and GPT4. Moreover, self-generating knowledge provides additional performance boost over self-generating exemplars, demonstrating its usefulness for the challenging Codeforces task. With our prompting method, GPT3.5-turbo achieves competitive performance with GPT4, with a 15% Acc@1 compared to GPT4's 16% Acc@1.

Figure 3 (more complete version in §D.3) provides a qualitative example of GPT3.5-turbo output generated using our prompt. The knowledge and exemplars generated by GPT3.5-turbo are indeed relevant to

| Prompting Method | (← scale down) text-curie-001 | text-davinci-001 | text-davinci-002 | (scale up →) text-davinci-003 |
|---|---|---|---|---|
| 0-shot | 2% | 6% | 13% | 14% |
| 0-shot CoT | 2% | 6% | 22% | 50% |
| 5-shot (fixed) CoT | 2% | 10% | 43% | 54% |
| 5-shot retrieved CoT | **3%** | **11%** | 47% | 57% |
| **Ours: Self-generated Exemplars** | 2% | 9% | **48%** | **61%** |

Table 4: Performance analysis using GSM8K task. **Across varied scales/strengths of base LLMs** (increasing from left to right), our prompting method outperforms 0-shot CoT and standard few-shot CoT with fixed exemplars. **Self-generated exemplars vs. retrieved exemplars**: our method, with self-generated exemplars, performs better with larger-scale LLMs, while few-shot CoT with retrieved exemplars performs better with smaller-scale LLMs.

the problem to solve, focusing on the prefix product algorithm. The final code generated by the LLM effectively applies the algorithm to solve the problem. In contrast, in the 0-shot CoT baseline, the LLM output does not recall relevant exemplars and fails to employ the prefix product algorithm, resulting in an incorrect solution (§D.3).

**BIG-Bench reasoning tasks.** Table 3 presents results for BIG-Bench tasks. Our prompting method outperforms baselines like 0-shot CoT, confirming its effectiveness across a wide range of tasks. Our method is also competitive with manual few-shot CoT. §D.4 offers GPT3.5-turbo output examples for the deductive reasoning task ("BIG-Bench formal fallacies"). Using our prompting method, the LLM generates relevant deductive reasoning exemplars. Conversely, 0-shot CoT, with no relevant exemplars, tends to adopt an incorrect approach to address the deductive reasoning problem.

## 6.2 KNOWLEDGE CAN COMPLEMENT EXEMPLARS

Generating knowledge alongside exemplars is particularly useful in Codeforces task (Table 2), where LLMs need to apply nontrivial algorithms for code generation. In our qualitative analysis, we observe two concrete advantages of generating knowledge: (1) knowledge act as high-level takeaways that complement low-level exemplars, which prevents LLMs from overly relying on specific exemplars and helps to generalize to new problems; (2) when generating knowledge, LLMs identify the core concepts of the problem and produce exemplars that align more closely in fundamental problem-solving approaches (e.g., the prefix product algorithm in Figure 3), rather than surface-level lexical similarities (e.g., without knowledge, LLMs tend to produce exemplars on palindromic sequences).

The performance gains achieved by generating knowledge are less significant in other tasks like GSM8K and BIG-Bench, however, likely because these tasks are less complex.

## 6.3 GENERATING VS RETRIEVING EXEMPLARS

A key motivation behind our idea of self-generating exemplars is its ability to offer relevant exemplars for problem solving. An alternative approach is to retrieve relevant exemplars from external data, provided there is a labeled dataset of exemplars (e.g., the training set of GSM8K, which includes solutions labeled with reasoning steps). What trade-offs exist between these two approaches?

The advantage of retrieval lies in its reliability. Exemplars retrieved from a labeled dataset are inherently valid and correct, unlike generated exemplars, which lack this guarantee. Nevertheless, retrieval typically needs labeled exemplars and involves the complexity of an additional retrieval step. The performance may also depend on the specific retriever used.

In contrast, generation is more self-contained and convenient, as it does not rely on external labeled data or retrieval steps. Additionally, generation may yield exemplars better tailored to specific test problems because it can draw upon the entire (pre-)training data the LLM has been exposed to. The downside of generation is that it may fail to produce valid exemplars if the LLMs are weak or have not learned problems related to the ones to be solved.

Table 4 shows empirical results for GSM8K task, comparing our self-generated exemplars method ("Ours") and the few-shot CoT method using exemplars retrieved from the GSM8K train set ("5-shot retrieved CoT"). We conducted experiments using base LLMs of various scales, from text-curie-001 to text-davinci-003, where scale broadly indicates the amount of training data and parameter count used by the LLM.

Our method outperforms the retrieved CoT with larger-scale LLMs, such as text-davinci-003. This is likely because the LLM has effectively learned related tasks during training and can generate useful

exemplars. Conversely, with smaller-scale LLMs, the retrieved CoT performs better, and self-generation fails to produce useful or valid exemplars.

## 6.4 SCALE OF BASE LLMs: ANALOGICAL PROMPTING EXCELS WITH LARGER MODELS

Table 4 presents the result of using varying scales and strengths of base LLMs, ranging from text-curie-001 to text-davinci-001 to text-davinci-002 and text-davinci-003 (more parameters and training data). Our prompting method surpasses vanilla 0-shot and 0-shot CoT across all scales. When using smaller-scale LLMs (text-curie-001 and text-davinci-001), few-shot CoT leveraging labeled exemplars exhibits superior performance compared to ours. However, as the LLMs are scaled up to text-davinci-002 and text-davinci-003, our method outperforms few-shot CoT. This is due to the LLMs' enhanced ability to self-generate more relevant and useful exemplars.

## 6.5 NUMBER OF EXEMPLARS TO GENERATE

In Table 5, we analyze the effect of varying the number of self-generated exemplars ($K$) in our approach. When $K = 1$, the LLM underperforms due to excessive reliance on a single exemplar generated. When $K \geq 3$, the LLM demonstrates consistent performance. This observation aligns with the findings in the standard few-shot in-context learning in LLMs (Brown et al., 2020).

| # Exemplars to self-generate | GSM8K | MATH |
|---|---|---|
| $K = 1$ | 76.1 | 34.8 |
| $K = 2$ | 77.0 | 36.7 |
| $K = 3$ | 77.5 | **37.3** |
| $K = 4$ | 77.3 | 37.0 |
| $K = 5$ | **77.8** | 37.1 |

Table 5: Analyzing the effect of varying the number of self-generated exemplars ($K$) in our approach. We assess performance on GSM8K and MATH tasks using GPT3.5-turbo as the base LLM.

## 6.6 QUALITATIVE ANALYSIS

We manually analyzed the performance of our prompting approach, based on a random sample of 50 correctly and 50 incorrectly solved problems from GSM8K + MATH (50%, 50%).

50 correctly solved problems:

- (6/50) Generated exemplars are irrelevant
- (9/50) Generated exemplars are relevant but contain incorrect solutions
- (35/50) Generated exemplars are relevant and correct

50 incorrectly solved problems:

- (10/50) Generated exemplars are irrelevant
- (12/50) Generated exemplars are relevant but contain incorrect solutions
- (28/50) Generated exemplars are relevant and correct, but LLM fails to solve the new problem:
    - (12/50) A generalization gap between the exemplars and the new problem
    - (8/50) Overreliance on specific exemplars, leading to misdirection
    - (8/50) Other issues, such as calculation errors

The generated exemplars were often relevant or correct. A common failure occurred when the LLM could not solve the new problem due to a generalization gap (e.g., the new problem is harder than the exemplars). This observation motivates future research to generate exemplars that not only possess relevance but also facilitate generalization for solving new problems.

## 7 CONCLUSION

We introduced *analogical prompting*, a new language model prompting approach that self-generates relevant reasoning exemplars for solving problems. This approach provides detailed, customized exemplars for individual problems without requiring labeled data, effectively addressing the challenges faced by existing 0-shot CoT and few-shot CoT prompting methods. Experimental results show that our approach outperforms 0-shot CoT and few-shot CoT in various reasoning tasks, including math problem solving, code generation, and other logical/temporal reasoning tasks.

# 8 LIMITATIONS AND FUTURE RESEARCH

One limitation of our approach is increased inference computation, as our approach generates more tokens than vanilla 0-shot and 0-shot CoT prompting. Compared to few-shot CoT, we use fewer input tokens and more output tokens, as exemplars are counted as input in few-shot CoT and as output in our approach.

Another limitation is that self-generation can fail if the LLM lacks sufficient strength or has not learned relevant knowledge to the new problems to solve. Conversely, with a stronger LLM, it can draw upon relevant prior knowledge to tackle slightly more complex problems. Therefore, our approach is better suited for stronger or larger-scale LLMs.

Finally, it is known that LLM performance can be influenced by specific prompt phrases used to query the model (Jiang et al., 2020), and our work is also subject to this prompt sensitivity.

## ACKNOWLEDGMENTS

We thank Qian Huang, Chen Liang, Andrew Dai and other colleagues at Google DeepMind, Stanford SNAP and P-Lambda groups for valuable feedback on this work.

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

# A   ADDITIONAL RELATED WORKS

## A.1   LANGUAGE MODELS AND REASONING

Reasoning involves the application of knowledge to derive solutions for new problems, often through a series of steps. Teaching language models to reason has been a long-standing area of research (Bottou, 2014; Zhao et al., 2023; Wei et al., 2022b).

To assess the reasoning capabilities of language models, researchers have created datasets for various tasks that demand reasoning skills. These tasks include multi-step question answering (Yang et al., 2018; Dua et al., 2019; Talmor et al., 2018), mathematical problem-solving (Cobbe et al., 2021; Hendrycks et al., 2021b), and code generation (Yu et al., 2018; Chen et al., 2021; Hendrycks et al., 2021a; Austin et al., 2021). In this study, we evaluate our methods using these diverse datasets.

To teach language models to reason effectively, one line of approaches involve training or fine-tuning them. This can include using reasoning-intensive data during training (Wu et al., 2021; Yasunaga et al., 2022b; Lightman et al., 2023; Moor et al., 2023), retrieving structured knowledge (Lin et al., 2019; Feng et al., 2020; Zhang et al., 2022a; Yasunaga et al., 2021; 2022a; Xie et al., 2022), and incorporating external modules for reasoning such as logic and program execution (Chen et al., 2018; 2019; Yasunaga & Liang, 2020; Gupta et al., 2020; Ren et al., 2021; Zhang et al., 2023).

Recently, with the rise of large language models (LLMs), prompting them to engage in reasoning has proven effective and gained attention. A common approach is prompting LLMs to generate intermediate reasoning steps, as demonstrated by the chain-of-thought method (Wei et al., 2022b; Kojima et al., 2022; Zhou et al., 2022; Wang et al., 2022), which assists LLMs in tackling complex reasoning tasks. Several studies have extended this approach with more structured algorithms and search methods (Khot et al., 2022; Drozdov et al., 2022; Zelikman et al., 2022; Yao et al., 2023; Press et al., 2022; Khattab et al., 2022; Jung et al., 2022), as well as longer-horizon action and planning (Yao et al., 2022; Hao et al., 2023; Park et al., 2023). Another line of work incorporates tools and programs into the prompting process to facilitate reasoning (Chen et al., 2023; 2022; Cai et al., 2023; Cheng et al., 2022; Kim et al., 2023; Zhou et al., 2023; Schick et al., 2023).

Our work complements these efforts to enhance LLM reasoning and is the first to draw inspiration from human analogical reasoning to improve LLM prompting.

## A.2   ANALOGICAL REASONING

Analogical reasoning is a cognitive process in which humans recall relevant past experiences when facing new challenges (Gentner & Holyoak, 1997; Gentner, 1983; Holyoak, 2012). This phenomenon has been studied extensively in psychology, revealing its significance in various cognitive tasks such as problem-solving (Gentner & Markman, 1997) and creativity (Ward et al., 1997). It is rooted in the capacity to identify structural and relational similarities between past and current situations, facilitating knowledge transfer (Dunbar, 2001).

Analogical reasoning has also influenced the development of artificial intelligence and machine learning algorithms (Carbonell, 1983; Mitchell et al., 1986) and has been employed as a reasoning benchmark for assessing machine learning models (Bian et al., 2014; Huang et al., 2021). A recent work also evaluates the ability of language models to identify analogies (Webb et al., 2023; Hu et al., 2023).

Our work makes a pioneering effort of applying analogical reasoning principles to enhance language model inference.

# B  CODEFORCES DATA COLLECTION

We scraped data from codeforces.com, following the procedure in prior works (Li et al., 2022b; Kulal et al., 2019; Yasunaga & Liang, 2021). We use Level-A problems that were published between January 2023 and August 2023. Each problem includes the full problem descriptions and test cases accessible on the website. The test cases include the public test cases found on the problem page and hidden test cases made available on the evaluation result pages once a contest is finished. Some of the hidden test cases were truncated on the website due to excessive input/output length, and we skipped those. We retained problems whose problem descriptions were within the length of 2000 tokens in GPT3.5-turbo, resulting in 50 problems. Because this dataset is relatively small, we conduct the evaluation twice and then report the average results.

# C  ADDITIONAL RESULTS

| Prompting Method | GSM8K | MATH |
|---|---|---|
| Ours: Non-diverse exemplars | 75.9% | 35.2% |
| Ours: Diverse exemplars | **77.8%** | **37.3%** |

Table 6: Ablation study on the effect of encouraging diversity in self-generating exemplars (§4.1).

| Prompting Method | Acc@1 | Acc@10 |
|---|---|---|
| Ours: Self-generate knowledge **after** exemplars | 14% | 27% |
| Ours: Self-generate knowledge **before** exemplars | **15%** | **29%** |

Table 7: Ablation study on the Codeforces task using GPT3.5-turbo. Self-generating knowledge before exemplars performs slightly better than self-generating knowledge after exemplars (§4.2).

| Prompting Method | GSM8K | MATH |
|---|---|---|
| Ours | 77.8% | 37.3% |
| Ours + Self-consistency | **85.3%** | **46.0%** |

Table 8: Self-consistency can be used to further boost the performance of our analogical prompting approach.

# D  FULL EXAMPLES

## D.1  MATH

Our prompt (self-generate exemplars):

```
Your task is to tackle mathematical problems. When presented with a math
    problem, recall relevant problems as examples. Afterward, proceed to
    solve the initial problem.

# Initial Problem:
An airline serves a dinner to all the passengers on an airplane. They get
    their choice of steak or fish. Three steak meals and three fish
    meals are set aside for the six-member crew. If the meals are
    distributed to the crew members randomly, what is the probability
    that both pilots get the fish?

# Instructions:
Make sure to include all of the following points:

## Relevant Problems:
Recall three examples of math problems that are relevant to the initial
    problem. Note that your problems should be distinct from each other
    and from the initial problem (e.g., involving different numbers and
    names). For each problem:
- After "Q: ", describe the problem
```

```
- After "A: ", explain the solution and enclose the ultimate answer in \
    boxed{}.

## Solve the Initial Problem:
Say "Let's solve the following math problem." Then formulate your
    response in the following format:
Q: Copy and paste the initial problem here.
A: Explain the solution and enclose the ultimate answer in \boxed{} here.
```

Output by GPT3.5-turbo:

```
Relevant Problems:
Q: In a bag, there are 5 red balls and 3 blue balls. If two balls are
    drawn at random without replacement, what is the probability that
    both balls are red?
A: The total number of balls is 8. The probability of drawing a red ball
    on the first draw is 5/8. After one red ball is drawn, there are 4
    red balls left out of 7 total balls. Therefore, the probability of
    drawing a second red ball is 4/7. The probability of both events
    happening is the product of the individual probabilities, so the
    probability of drawing two red balls is (5/8) * (4/7) = \boxed{\frac
    {5}{14}}.

Q: A box contains 10 red marbles and 5 blue marbles. If three marbles are
     drawn at random without replacement, what is the probability that
    all three marbles are red?
A: The total number of marbles is 15. The probability of drawing a red
    marble on the first draw is 10/15. After one red marble is drawn,
    there are 9 red marbles left out of 14 total marbles. Therefore, the
    probability of drawing a second red marble is 9/14. After two red
    marbles are drawn, there are 8 red marbles left out of 13 total
    marbles. Therefore, the probability of drawing a third red marble is
    8/13. The probability of all three events happening is the product of
     the individual probabilities, so the probability of drawing three
    red marbles is (10/15) * (9/14) * (8/13) = \boxed{\frac{24}{91}}.

Q: A bag contains 4 green marbles, 3 red marbles, and 2 blue marbles. If
    two marbles are drawn at random without replacement, what is the
    probability that both marbles are green?
A: The total number of marbles is 9. The probability of drawing a green
    marble on the first draw is 4/9. After one green marble is drawn,
    there are 3 green marbles left out of 8 total marbles. Therefore, the
     probability of drawing a second green marble is 3/8. The probability
     of both events happening is the product of the individual
    probabilities, so the probability of drawing two green marbles is
    (4/9) * (3/8) = \boxed{\frac{1}{6}}.

Let's solve the following math problem:
An airline serves a dinner to all the passengers on an airplane. They get
     their choice of steak or fish. Three steak meals and three fish
    meals are set aside for the six-member crew. If the meals are
    distributed to the crew members randomly, what is the probability
    that both pilots get the fish?
A: There are a total of 6 crew members, so the probability of the first
    pilot getting the fish is 3/6. After one fish meal is given to the
    first pilot, there are 2 fish meals left out of 5 total meals.
    Therefore, the probability of the second pilot getting the fish is
    2/5. The probability of both events happening is the product of the
    individual probabilities, so the probability that both pilots get the
     fish is (3/6) * (2/5) = \boxed{\frac{1}{5}}.
```

## D.2  GSM8K

Our prompt (self-generate exemplars):

```
Your task is to tackle mathematical problems. When presented with a math
    problem, recall relevant problems as examples. Afterward, proceed to
    solve the initial problem.

# Initial Problem:
Kyle bought last year's best-selling book for $19.50. This is with a 25%
    discount from the original price. What was the original price of the
    book?

# Instructions:
Make sure that your response follows the instructions below.

## Analogous Problems:
Offer five diverse examples of math problems that are relevant or
    analogous to the initial problem. For each problem, elaborate on the
    solution and conclude with the ultimate answer (enclosed in \boxed{})
    . For each problem:
- After "Q: ", describe the problem
- After "A: ", explain the solution and enclose the ultimate answer in \
    boxed{}.

## Solve the Initial Problem:
Q: Copy and paste the initial problem here.
A: Explain the solution and enclose the ultimate answer in \boxed{} here.
```

Output by GPT3.5-turbo:

```
Analogous Problems:

Q1: A shirt is on sale for 20% off the original price of $40. What is the
     sale price of the shirt?
A: The original price of the shirt is $40. With a 20% discount, the
    discount amount is 20% of $40, which is 0.20 * $40 = $8. Therefore,
    the sale price of the shirt is $40 - $8 = $32. The answer is $\boxed
    {32}$.

Q2: A store is offering a 15% discount on all items. If a customer buys a
     pair of shoes for $85, what was the original price of the shoes?
A: Let's assume the original price of the shoes is $x. With a 15%
    discount, the discount amount is 15% of $x, which is 0.15 * $x = $0
    .15x. The sale price of the shoes is $85, so we can set up the
    equation: $x - $0.15x = $85. Simplifying the equation, we get 0.85x =
     $85. Dividing both sides by 0.85, we find that x = $100. Therefore,
    the original price of the shoes was $\boxed{100}$.

Q3: A restaurant offers a 10% discount on all meals. If a customer pays
    $45 for a meal, what was the original price of the meal?
A: Let's assume the original price of the meal is $x. With a 10% discount
    , the discount amount is 10% of $x, which is 0.10 * $x = $0.10x. The
    sale price of the meal is $45, so we can set up the equation: $x - $0
    .10x = $45. Simplifying the equation, we get 0.90x = $45. Dividing
    both sides by 0.90, we find that x = $50. Therefore, the original
    price of the meal was $\boxed{50}$.

Q4: A laptop is on sale for 30% off the original price of $800. What is
    the sale price of the laptop?
A: The original price of the laptop is $800. With a 30% discount, the
    discount amount is 30% of $800, which is 0.30 * $800 = $240.
    Therefore, the sale price of the laptop is $800 - $240 = $560. The
    answer is $\boxed{560}$.
```

```
Q5: A toy store is offering a 12% discount on all toys. If a customer
    buys a toy for $22, what was the original price of the toy?
A: Let's assume the original price of the toy is $x. With a 12% discount,
    the discount amount is 12% of $x, which is 0.12 * $x = $0.12x. The
    sale price of the toy is $22, so we can set up the equation: $x - $0
    .12x = $22. Simplifying the equation, we get 0.88x = $22. Dividing
    both sides by 0.88, we find that x = $25. Therefore, the original
    price of the toy was $\boxed{25}$.

Solve the Initial Problem:

Q: Kyle bought last year's best-selling book for $19.50. This is with a
    25% discount from the original price. What was the original price of
    the book?
A: Let's assume the original price of the book is $x. With a 25% discount
    , the discount amount is 25% of $x, which is 0.25 * $x = $0.25x. The
    sale price of the book is $19.50, so we can set up the equation: $x -
     $0.25x = $19.50. Simplifying the equation, we get 0.75x = $19.50.
    Dividing both sides by 0.75, we find that x = $26. Therefore, the
    original price of the book was $\boxed{26}$.
```

Our prompt (self-generate exemplars) with in-context demonstration, used by text-davinci models:

```
Your task is to solve math problems.

For example,

# Problem:
Q: A pet shelter spends $30 on grooming for each cat, $60 for each adult
    dog, and $90 for each puppy to prepare them for adoption. If they
    prepare 3 cats, 2 adult dogs, and 4 puppies for adoption, what is the
     total cost of getting them ready?

# Relevant problems:
Q: In a bakery, there is a sale of baked goods. Every muffin costs $2,
    every cookie $1, and a loaf of bread $3. How much do you need to pay
    for five muffins, six cookies, and two loaves of bread?
A: Let's calculate the cost for each type of item:
Five muffins cost: 5 * $2 = $10.
Six cookies cost: 6 * $1 = $6.
Two loaves of bread cost: 2 * $3 = $6.
Now, add up the costs for each item to find the total cost:
Total cost = $10 (muffins) + $6 (cookies) + $6 (bread) = $22.
So, for five muffins, six cookies, and two loaves of bread, you would
    need to pay \\boxed{{$22}}.

Q: In a garden, there are four different types of fruit trees. The first
    type of tree has 15 apples, the second type has 10 oranges, the third
     type has 24 pears, and the fourth type has 30 plums. Sarah decides
    to pick fruit from these trees for making a fruit salad. She picks
    40% of the apples, 60% of the oranges, 75% of the pears, and 50% of
    the plums. How many pieces of fruit are in her fruit salad?
A: Number of apples Sarah picks: 15 x 40% = 6 apples.
Number of oranges Sarah picks: 10 x 60% = 6 oranges.
Number of pears Sarah picks: 24 x 75% = 18 pears.
Number of plums Sarah picks: 30 x 50% = 15 plums.
Now, let's calculate the total number of pieces of fruit in her fruit
    salad:
Total = 6 apples + 6 oranges + 18 pears + 15 plums = 6 + 6 + 18 + 15 = 45
     pieces of fruit.
So, there are \\boxed{{45}} pieces of fruit in Sarah's fruit salad.
```

```
Q: Sarah runs a small jewelry business and sells necklaces for 25% more
    than it costs to make them. If she sold 40 necklaces and earned $400,
    how much does each necklace cost to make?
A: To find out how much each necklace costs to make, we can start by
    determining the selling price per necklace.
First, we need to calculate the selling price per necklace. Since Sarah
    sold 40 necklaces and earned $400, we can use the formula:
Selling Price per Necklace = $400 / 40 = $10
Now that we know each necklace was sold for $10, we can find out the cost
    to make each necklace. Sarah sells them for 25% more than their cost
    .
Cost to Make each Necklace = $10 / (1 + 0.25) = $10 / 1.25 = $8
So, each necklace costs \\boxed{{$8}} to make.

Q: A machine at a coffee shop can usually brew a pot of coffee in 6
    minutes. However, this morning a malfunction occurred, and it took 10
     minutes to brew one pot of coffee. If the malfunction remains
    unresolved, how much longer will it take to brew 8 pots of coffee
    than on a normal day when the machine works fine?
A: On a regular day, it would take 6 * 8 = 48 minutes to brew 8 pots of
    coffee.
With the malfunction, it will take 10 * 8 = 80 minutes to brew 8 pots of
    coffee.
It will take 80 - 48 = \\boxed{{32}} more minutes if the malfunction goes
    unresolved.

Q: In a jar, there were 800 candies. Sarah took 3/8 of the total number
    of candies, while David ate 2/5 of the remaining candies. What
    percentage of the original candies remained?
A: Sarah took 3/8 of the total candies, which is 3/8 * 800 = 300 candies.
After Sarah's candy-taking, there were 800 - 300 = 500 candies left.
David then ate 2/5 of the remaining candies, which is 2/5 * 500 = 200
    candies.
After David's candy-eating, there were 500 - 200 = 300 candies left.
The percentage of candies that remained out of the original amount is
    (300 / 800) * 100% = \\boxed{{37.5}}%.

# Solve the initial problem step by step:
Q: A pet shelter spends $30 on grooming for each cat, $60 for each adult
    dog, and $90 for each puppy to prepare them for adoption. If they
    prepare 3 cats, 2 adult dogs, and 4 puppies for adoption, what is the
     total cost of getting them ready?
A: To find the total cost of getting these animals ready for adoption, we
    'll multiply the cost of each type of animal by the number of that
    type being prepared:
Cost for 3 cats = 3 * $30 = $90
Cost for 2 adult dogs = 2 * $60 = $120
Cost for 4 puppies = 4 * $90 = $360
Now, add up the costs for each type:
Total cost = $90 + $120 + $360 = $570
So, the total cost to get 3 cats, 2 adult dogs, and 4 puppies ready for
    adoption is \\boxed{{$570}}.

Solve the following math problem step by step.

# Problem:
Q: Kyle bought last year's best-selling book for $19.50. This is with a
    25% discount from the original price. What was the original price of
    the book?

# Relevant problems:
```

### D.3    CODEFORCES

Our prompt (self-generate exemplars and knowledge):

```
Your goal is to write Python3 code to solve competitive programming
    problems. Given a problem, explain the core concepts in it and
    provide other relevant problems. Then solve the original problem.

# Problem:
You are given a sequence a_1, a_2, \ldots, a_n. Each element of a is 1 or
     2. Find out if an integer k exists so that the following conditions
     are met.
  1 \leq k \leq n-1, and  a_1 \cdot a_2 \cdot \ldots \cdot a_k = a_{k+1}
       \cdot a_{k+2} \cdot \ldots \cdot a_n, where \cdot denotes a product
      .
If there exist multiple k that satisfy the given condition, print the
    smallest.

Input
Each test contains multiple test cases. The first line contains the
    number of test cases t (1 \le t \le 100). Description of the test
    cases follows.
The first line of each test case contains one integer n (2 \leq n \leq
    1000).
The second line of each test case contains n integers a_1, a_2, \ldots,
    a_n (1 \leq a_i \leq 2).

Output
For each test case, if there is no such k, print -1.
Otherwise, print the smallest possible k.

# Instruction:

## Algorithms:
Identify the core concepts or algorithms used to solve the problem.

## Tutorial:
Write a useful tutorial about these algorithms.

## Example Problems:
Provide three examples of relevant competitive programming problems that
    involve these algorithms. For each problem, describe the problem,
    explain the solution in detail, and then write the correct Python3
    code.

## Python3 code to solve the original problem:
Include the following points in your response:
- Explanation of the solution:
- Python3 code to solve the problem:
```

Output by GPT3.5-turbo-16k:

```
## Algorithms:
The core algorithm for solving this problem is the prefix product
    algorithm.

## Tutorial:
The prefix product algorithm is used to calculate the product of all the
    elements in a prefix of an array. It is a useful algorithm for
    solving problems that involve calculating the product of all the
    elements in a subarray.

The algorithm works by initializing a variable "prefix" to 1, and then
    iterating through the array from left to right. At each iteration,
```

the current element is multiplied with the prefix, and the result is
    stored in the prefix variable. This way, the prefix variable keeps
    track of the product of all the elements in the prefix of the array.

Here is the pseudocode for the prefix product algorithm:

```
prefix = 1
for i in range(n):
    prefix = prefix * arr[i]
```

The time complexity of this algorithm is O(n), where n is the size of the
    input array.

## Example Problems:
1. Problem: Given an array of integers, find the product of all the
   elements in each subarray of size k.
   Solution: We can use the prefix product algorithm to solve this
       problem. We initialize a prefix variable to 1, and then iterate
       through the array from left to right. At each iteration, we
       multiply the current element with the prefix, and store the result
        in the prefix variable. If the index i is greater than or equal
       to k-1, we divide the prefix by the element arr[i-k+1], which
       represents the first element of the subarray of size k starting at
        index i-k+1. The result is the product of all the elements in the
        subarray of size k starting at index i-k+1. We can store these
       results in a new array and return it as the output.

```python
def product_of_subarrays(arr, k):
    n = len(arr)
    prefix = 1
    result = []

    for i in range(n):
        prefix = prefix * arr[i]

        if i >= k-1:
            result.append(prefix)
            prefix = prefix // arr[i-k+1]

    return result
```

2. Problem: Given an array of integers, find the longest subarray with a
   product less than or equal to a given target.
   Solution: We can use the prefix product algorithm to solve this
       problem. We initialize a prefix variable to 1, and then iterate
       through the array from left to right. At each iteration, we
       multiply the current element with the prefix, and store the result
        in the prefix variable. If the prefix becomes greater than the
       target, we divide the prefix by the first element of the subarray,
        and move the start pointer of the subarray one step forward. We
       update the maximum length of the subarray if necessary.

```python
def longest_subarray_with_product_less_than_target(arr, target):
    n = len(arr)
    prefix = 1
    start = 0
    max_length = 0

    for i in range(n):
        prefix = prefix * arr[i]
```

```
        while prefix > target and start <= i:
            prefix = prefix // arr[start]
            start += 1

        max_length = max(max_length, i-start+1)

    return max_length
```

3. Problem: Given an array of integers, find the smallest k such that the
     product of all the elements in the prefix of the array is equal to
    the product of all the elements in the suffix of the array.
   Solution: We can use the prefix product algorithm to solve this
       problem. We initialize a prefix variable to 1, and then iterate
       through the array from left to right. At each iteration, we
       multiply the current element with the prefix, and store the result
        in the prefix variable. We also initialize a suffix variable to
       1, and then iterate through the array from right to left. At each
       iteration, we multiply the current element with the suffix, and
       store the result in the suffix variable. If the prefix is equal to
        the suffix, we return the current index as the smallest k. If no
       such index is found, we return -1.

```python
def smallest_k_with_equal_product(arr):
    n = len(arr)
    prefix = 1
    suffix = 1

    for i in range(n):
        prefix = prefix * arr[i]
        suffix = suffix * arr[n-i-1]

        if prefix == suffix:
            return i+1

    return -1
```

## Python3 code to solve the original problem:
Explanation of the solution:
- Read the number of test cases.
- Iterate through each test case.
  - Read the number of elements in the sequence and the sequence itself.
  - Initialize prefix and suffix variables to 1, and k to -1.
  - Calculate the product of all the elements in the sequence using the
     prefix product algorithm.
  - Iterate through the sequence from right to left.
    - Update the prefix and suffix variables.
    - If the prefix is equal to the suffix, update k to the current index
       .
  - Print the value of k.

```python
# Read the number of test cases
t = int(input())

# Process each test case
for _ in range(t):
    # Read the test case input
    n = int(input())
    arr = list(map(int, input().split()))

    # Initialize prefix and suffix variables
```

```
        prefix = 1
        suffix = 1
        k = -1

        # Calculate prefix product
        for i in range(n):
            prefix = prefix * arr[i]

        # Calculate suffix product and find smallest k
        for i in range(n-1, 0, -1):
            prefix = prefix // arr[i]
            suffix = suffix * arr[i]

            if prefix == suffix:
                k = i

    # Print the result
    print(k)
```

Below is a baseline output generated by GPT3.5-turbo-16k using vanilla 0-shot prompt. This output does not recall any relevant exemplars and fails to employ the prefix product algorithm, resulting in an incorrect solution.

```
def smallest_k(t, test_cases):
    results = []
    for i in range(t):
        n = test_cases[i][0]
        a = test_cases[i][1:]
        k = -1
        for j in range(1, n):
            if a[:j] == a[j:]:
                k = j
                break
        results.append(k)
    return results

t = int(input())
test_cases = []
for _ in range(t):
    n = int(input())
    a = list(map(int, input().split()))
    test_cases.append([n] + a)

output = smallest_k(t, test_cases)
for o in output:
    print(o)
```

## D.4   BIG-BENCH: FORMAL FALLACIES

Our prompt (self-generate exemplars):

```
Your task is to tackle reasoning problems. When presented with a problem,
    recall relevant problems as examples. Afterward, proceed to solve
    the initial problem.

# Initial Problem:
"Is Siri a stepsister of Mary? Is Susan related to Kate? In large
    families, it is sometimes difficult to keep track of all one's
    relatives. The following argument seeks to clarify some such
    relations: To start with, everyone who is a schoolmate of Mildred is
    a daughter of Jeanette, too. Now, being a daughter of Jeanette is
    sufficient for not being a cousin of Cristi. Therefore, no schoolmate
```

```
      of Mildred is a cousin of Cristi." Is the argument, given the
   explicitly stated premises, deductively valid or invalid? Options: -
   valid  - invalid

# Instructions:
Make sure to include all of the following points:

## Relevant Problems:
Recall three examples of problems that are relevant to the initial
   problem. Note that your problems must be distinct from each other and
    from the initial problem. For each problem:
- After "Q: ", describe the problem
- After "A: ", explain the solution and enclose the ultimate answer in \
   boxed{}.

## Solve the Initial Problem:
Say "Let's solve the following reasoning problem." Then formulate your
   response in the following format:
Q: Copy and paste the initial problem here.
A: Explain the solution and enclose the ultimate answer in \boxed{} here.
```

Output by GPT3.5-turbo:

```
Relevant Problems:
Q: In a certain town, all doctors are required to have a medical degree.
   John is a doctor. Therefore, John must have a medical degree. Is this
    argument valid or invalid?
A: This argument is valid because it follows the logical form of a
   categorical syllogism, where the first premise establishes a
   universal statement and the second premise applies that statement to
   a specific case. Therefore, the conclusion can be logically deduced.
   \boxed{valid}

Q: All cats have fur. Fluffy is a cat. Therefore, Fluffy has fur. Is this
    argument valid or invalid?
A: This argument is valid because it follows the logical form of a
   categorical syllogism, where the first premise establishes a
   universal statement and the second premise applies that statement to
   a specific case. Therefore, the conclusion can be logically deduced.
   \boxed{valid}

Q: If it is raining, then the ground is wet. The ground is wet. Therefore
   , it must be raining. Is this argument valid or invalid?
A: This argument is invalid because it commits the fallacy of affirming
   the consequent. Just because the ground is wet does not necessarily
   mean that it is raining, as there could be other factors that caused
   the ground to become wet. \boxed{invalid}

Solve the Initial Problem:
Let's solve the following reasoning problem:

Q: "Is Siri a stepsister of Mary? Is Susan related to Kate? In large
   families, it is sometimes difficult to keep track of all one's
   relatives. The following argument seeks to clarify some such
   relations: To start with, everyone who is a schoolmate of Mildred is
   a daughter of Jeanette, too. Now, being a daughter of Jeanette is
   sufficient for not being a cousin of Cristi. Therefore, no schoolmate
    of Mildred is a cousin of Cristi." Is the argument, given the
   explicitly stated premises, deductively valid or invalid?
A: The argument is deductively valid. The premises state that everyone
   who is a schoolmate of Mildred is a daughter of Jeanette, and being a
    daughter of Jeanette is sufficient for not being a cousin of Cristi.
    Therefore, it logically follows that no schoolmate of Mildred is a
   cousin of Cristi. \boxed{valid}
```

Below is a baseline output generated by GPT3.5-turbo using 0-shot CoT prompt. This output does not recall any relevant exemplars and adopts an incorrect approach to address the deductive reasoning problem.

```
The argument is deductively invalid.

To determine the validity of the argument, we need to analyze the logical
    structure and see if the conclusion necessarily follows from the
    premises.

Let's break down the argument:

1. Everyone who is a schoolmate of Mildred is a daughter of Jeanette.
2. Being a daughter of Jeanette is sufficient for not being a cousin of
    Cristi.

Conclusion: No schoolmate of Mildred is a cousin of Cristi.

The first premise establishes a relationship between being a schoolmate
    of Mildred and being a daughter of Jeanette. However, it does not
    provide any information about the relationship between being a
    schoolmate of Mildred and being a cousin of Cristi.

The second premise establishes a relationship between being a daughter of
    Jeanette and not being a cousin of Cristi. However, it does not
    provide any information about the relationship between being a
    schoolmate of Mildred and being a cousin of Cristi.

Therefore, the conclusion that no schoolmate of Mildred is a cousin of
    Cristi cannot be logically derived from the given premises. The
    argument is deductively invalid.

\boxed{invalid}
```

