# OpenReview forum: "Large Language Models as Analogical Reasoners"
_ICLR.cc/2024/Conference — ICLR 2024 poster_

### Official Review · Reviewer_ey56 · 2023-10-24

**Soundness:** 3 good
**Presentation:** 3 good
**Contribution:** 3 good
**Rating:** 8
**Confidence:** 4

**Summary:**

This work presents a novel prompting method inspired by the concept of analogical reasoning in cognitive science. The method is shown to outperform few-shot chain-of-thought methods, while not requiring any few-shot examples.

**Strengths:**

- The proposed method involves an interesting take on concepts from cognitive science, and yields consistent improvements across reasoning benchmarks.
- The proposed method is simple to implement, and does not require any task-specific prompts or training examples.
- A qualitative analysis is performed of the generated exemplars and their relationship to downstream performance.
- The code dataset is limited to recently published problems, thus addressing test set contamination concerns.

**Weaknesses:**

- The paper emphasizes the importance of instructing the LLM to generate *distinct* exemplars, but there is no ablation performed for this specific aspect of the method.
- It seems unlikely that the generated exemplars are literally retrieved from memory in the sense that they are in human reasoning. It seems more likely that these are novel problems generated by the LLM, based on general statistical knowledge. I don't think this really undermines the usefulness of the approach, but it might be worthwhile to briefly discuss this issue.
- The caption for Table 1 mentions that an in-context demonstration was used for the davinci models, but I couldn't find any explanation of this description (e.g., does this include an in-context training example, or merely a demonstration of the formatting?).

Minor comments:
- The authors might consider citing work that analyzes the analogical reasoning ability of LLMs [1,2] (though I should note that I don't think this undermines the contribution of the present work). There are also a few references that would be good to include when introducing the general concept of analogical reasoning and its role in the psychology literature [3,4].

[1] Webb, T., Holyoak, K. J., & Lu, H. (2023). Emergent analogical reasoning in large language models. Nature Human Behaviour.

[2] Hu, X., Storks, S., Lewis, R. L., & Chai, J. (2023). In-Context Analogical Reasoning with Pre-Trained Language Models. arXiv preprint arXiv:2305.17626.

[3] Gentner, D. (1983). Structure-mapping: A theoretical framework for analogy. Cognitive science, 7(2), 155-170.

[4] Holyoak, K. J. (2012). Analogy and relational reasoning. The Oxford handbook of thinking and reasoning, 234-259.

**Questions:**

I am curious to hear the authors thoughts regarding whether the generated exemplars are genuinely retrieved from memory, or are novel problems based on general statistical knowledge. The latter case seems more consistent with the memory mechanisms in LLMs (and with their tendency to generate fabricated but plausible sounding information), but it is not something that has been considered much in the psychology literature on analogical reasoning, and it is somewhat more difficult to understand how it could improve performance.

---

> ### Author Response · Authors · 2023-11-18
> **Thanks for your review, and our response**
>
> We sincerely thank the reviewer for insightful feedback. We have incorporated all suggestions in our paper. We thank the reviewer for describing that our work offers a novel, interesting, and general-purpose prompting method and conducts careful experiments that mitigate test set contamination. We respond to the reviewer’s concerns and questions below.
>
> > no ablation performed for generating distinct exemplars
>
> Thank you for pointing this out. Following is the ablation result on instructing the LLM to generate *distinct* exemplars or not:
> - GSM8K non-distinct vs distinct: 75.9 vs 77.8
> - MATH non-distinct vs distinct: 35.2 vs 37.3
>
>
> > It seems unlikely that the generated exemplars are literally retrieved from memory in the sense that they are in human reasoning. It seems more likely that these are novel problems generated by the LLM, based on general statistical knowledge. I don't think this really undermines the usefulness of the approach, but it might be worthwhile to briefly discuss this issue. / I am curious to hear the authors thoughts regarding whether the generated exemplars are genuinely retrieved from memory, or are novel problems based on general statistical knowledge.
>
> This is an excellent point and you are correct. We observed that some of the self-generated exemplars resemble the potential training data of the LLM (e.g., GSM8K training set, Geeksforgeeks coding problems), but many of the exemplars are newly generated and appear to be an interpolation of problems seen by the LLM that are related to the target problem. We find this phenomenon very intriguing, suggesting that LLMs may be capable of generating useful exemplars beyond the existent/seen exemplars.
>
>
> > The caption for Table 1 mentions that an in-context demonstration was used for the davinci models, but I couldn't find any explanation of this description.
>
> Thank you for pointing this out. We added the description of in-context demonstration in Appendix D.2. It provides an example of formatting the generation of exemplars and solution to the initial problem.
>
>
> > The authors might consider citing work that analyzes the analogical reasoning ability of LLMs [1,2] (though I should note that I don't think this undermines the contribution of the present work).
>
> Thank you very much for pointing us to the related works. We have added them to our manuscript.

---

> > ### Comment · Reviewer_ey56 · 2023-11-22
> > **Response to rebuttal**
> >
> > Thank you to the authors for these responses and clarifications. I enthusiastically support the paper's acceptance.

---

> > > ### Author Response · Authors · 2023-11-23
> > > **Thanks for your strong support!**
> > >
> > > Thank you very much for reading our response and supporting our work. We really appreciate it!

---

### Official Review · Reviewer_fGoM · 2023-10-31

**Soundness:** 3 good
**Presentation:** 3 good
**Contribution:** 2 fair
**Rating:** 5
**Confidence:** 5

**Summary:**

This paper introduces the analogical prompting method. It's a method similar to CoT, but instead of just plaining "let's think step by step", it first prompts the model to automatically generate some knowledge and exemplars for solving tasks. Experiments on mathematical reasoning, code generation, and big-bench confirm its validity.

**Strengths:**

- This paper is well-written, with a detailed depiction of the design principles and implementation details for the proposed analogical prompting method. Experiment results on GSM8K, MATH, codeforces, and a bigbench subset show moderate improvement compared to few-shot CoT, and a noticeable boost compared to other zero-shot prompting methods.

- The analogical prompting method is well-motivated and intuitive enough to follow. Some ablation studies on the scalability, w/o knowledge, and the number of exemplars are good.

**Weaknesses:**

- Limited technical contribution. Although I think this paper is a good example of how to perform prompting engineering when solving zero-shot mathematical reasoning and code generation tasks, it is still more like a trick to an existing method (analogous to CoT->zero-shot CoT, in this case is retrival few-shot -> self-generated few-shot).

- The codeforces dataset only contains 50 questions; perhaps it is too small to make claims on the improvements (~2% is only one more solved question). Any experiments on larger code generation tasks, e.g., HumanEval?

**Questions:**

There is error analysis for incorrectly solved tasks, how about correctly solved questions? How many generated exemplars are wrong, but the solution to the new question is correct (i.e., it is known that sometimes LLMs can few-shot generalize from wrong exemplars)?

---

> ### Author Response · Authors · 2023-11-18
> **Thanks for your review, and our response**
>
> We sincerely thank the reviewer for constructive feedback. We have incorporated all suggestions in our paper. We are glad that the reviewer describes our work as well-motivated, having good design principles, and offering valuable ablation studies on ​​the scalability and knowledge, etc. We respond to the reviewer’s concerns and questions below.
>
> > Although I think this paper is a good example of how to perform prompting engineering when solving zero-shot mathematical reasoning and code generation tasks, it is still more like a trick to an existing method (analogous to CoT->zero-shot CoT, in this case is retrieval few-shot -> self-generated few-shot).
>
> Thank you for your valuable feedback. We would like to highlight that our approach makes significant methodological contributions, beyond prompt engineering or the transition from retrieval few-shot to self-generated few-shot:
> - **New LLM reasoning paradigm**: We introduce a new problem-solving paradigm for LLM, inspired by analogical reasoning in psychology. As Polya's book "How to Solve It" (2004) states, when solving a new problem, humans draw from how they solved related problems in the past. We are the first to show that this strategy aids LLMs in solving complex reasoning problems (Section 1).
> - **Design of new techniques**: To ground this strategy in LLM prompting, we analyzed the design space and present effective techniques:
>   - Diversity in generated exemplars (Section 4.1)
>   - Sequential vs independent generation of exemplars (Section 4.1)
>   - Generation of high-level knowledge in addition to low-level exemplars (Section 4.2)
> - **Impact of generation v.s. retrieval**: Our method, based on generation rather than retrieval, offers greater generality and flexibility. It unlocks the generation of various forms of relevant information, such as exemplars, knowledge, formulas, tutorials, etc. (as experimented in Section 4.2). This goes beyond traditional retrieval methods that were confined to a pre-defined set (e.g., GSM8K train set), and empirically yields superior results too (Table 4).
>
> > The codeforces dataset only contains 50 questions; perhaps it is too small to make claims on the improvements (~2% is only one more solved question). Any experiments on larger code generation tasks, e.g., HumanEval?
>
> This is a great point. The reason we used Codeforces 2023 for evaluation, despite its smaller size, was to prevent test set contamination; larger code generation datasets such as HuanEval and older versions of Codeforces may be used for LLM training, introducing the risk of contamination. To mitigate the issue of the small evaluation set, we repeated the experiments multiple times, each time using different LLM output samples, and then reported the average results (Section B).
>
> We also evaluated our method on the larger, older Codeforces dataset from AlphaCode, involving 300 coding problems. Our prompting method outperformed the baseline CoT prompting by 4% in this setting as well.
>
> > There is error analysis for incorrectly solved tasks, how about correctly solved questions? How many generated exemplars are wrong, but the solution to the new question is correct (i.e., it is known that sometimes LLMs can few-shot generalize from wrong exemplars)?
>
> Thank you for pointing this out. You are right. While not too often, there are cases where generated exemplars are wrong but the solution to the new question is correct. This result aligns with the previous finding that LLMs can few-shot generalize from wrong exemplars. Below are the details of 50 correctly solved problems and 50 incorrectly solved problems from GSM8K+MATH:
> - 50 correctly solved problems:
>   - (15/50) Generated exemplars are incorrect
>   - (35/50) Generated exemplars are correct
> - 50 incorrectly solved problems:
>   - (22/50) Generated exemplars are incorrect
>   - (28/50) Generated exemplars are correct

---

> > ### Comment · Reviewer_fGoM · 2023-11-21
> >
> > Thanks for the authors' response. After reading the new experiment results and discussions from reviewer ExxM, my concerns remain the same. It looks like the correctness of exemplars doesn't matter much, so it is technically not analogical reasoning.

---

> > > ### Author Response · Authors · 2023-11-22
> > > **Thanks for your comment, and our clarification**
> > >
> > > Thank you for reading our response!
> > >
> > > >  It looks like the correctness of exemplars doesn't matter much, so it is technically not analogical reasoning.
> > >
> > > The correctness of exemplars does matter in our evaluation result: when the exemplars are correct, the final problem-solving accuracy is indeed higher than when the exemplars are incorrect.
> > >
> > > It appears that the reviewer's underlying concerns are that (a) even when the exemplars are incorrect, the LLM sometimes correctly solves the target problem and (b) even when the exemplars are correct, the LLM sometimes fails to solve the target problem. Below, let us clarify that **these situations are completely reasonable** and do not have much to do with whether our method is analogical reasoning or not.
> > >
> > > > Regarding the situation (a) "even when the exemplars are incorrect, the LLM sometimes correctly solves the target problem":
> > >
> > > This is reasonable, as it is known that LLMs can few-shot learn even from wrong in-context exemplars (e.g., https://arxiv.org/abs/2202.12837). This is not related to whether our method is analogical reasoning or not.
> > >
> > > > Regarding the situation (b) "even when the exemplars are correct, the LLM sometimes fails to solve the target problem":
> > >
> > > This is also reasonable. **Few-shot CoT always uses correct exemplars, but can fail to solve the target problem too – in fact, it fails more often than our method (Table 1). Our main contribution is that we show our method outperforms manual few-shot CoT, i.e., LLMs benefit more from our self-generated exemplars than hand-written exemplars.**
> > >
> > > Additionally, reviewers might see our analysis result "among 50 unsolved problems, (22/50) Generated exemplars are incorrect; (28/50) Generated exemplars are correct" and might be concerned that "correct exemplars do not benefit solving the target problem correctly". We would like to respectfully explain that this is not the correct interpretation. **This analysis is an error analysis, which only looks at unsolved problems (only a part of the total event space) rather than many other solved problems, for which LLMs often benefit from correct exemplars.**
> > >
> > > Moreover, among unsolved problems, the error modes are always split into two cases: 1. generated exemplars are irrelevant or incorrect, and 2. generated exemplars are correct. If the LLM is strong, then it is good at generating relevant and correct exemplars, so naturally, Case 2 “LLM failed to benefit from correct exemplars” will be the majority (similarly, if the LLM is weak, then Case 1 will be the majority and Case 2 will be the minority). Therefore, **the relative frequency of correct exemplars vs incorrect exemplars among unsolved problems is not related to whether our method is analogical reasoning or not.**
> > >
> > > To conclude, we would like to clarify once again that our goal is not about achieving 100% problem-solving accuracy; CoT with correct exemplars does not achieve 100% accuracy either. **Our main contribution is that our method outperforms few-shot CoT, i.e., LLMs benefit more from our self-generated exemplars than hand-written exemplars.**
> > >
> > > Thank you for your consideration.

---

### Official Review · Reviewer_ExxM · 2023-11-01

**Soundness:** 3 good
**Presentation:** 3 good
**Contribution:** 3 good
**Rating:** 5
**Confidence:** 4

**Summary:**

The authors propose the use of analogical prompting to enhance reasoning performance. This method requires LLMs to first self-generate relevant examples or knowledge before attempting to solve a problem. Then, the model provides a response based on the generated concepts.

**Strengths:**

1. The paper is well-organized and easy to understand. And the proposed method is intuitive.
2. The experiments demonstrate the effectiveness of the proposed methods across diverse datasets, while the ablation study indicates that the method outperforms the baseline approaches.

**Weaknesses:**

1. Will the language model be easily distracted when the generated examples are irrelevant or incorrect? The experiments in section 6.6 should provide more details. For instance, it should specify how many problems in MATH fail due to incorrect and irrelevant generated examples.
2. Can current LLMs generate helpful examples for challenging questions? The authors are encouraged to include more examples in the paper.
3. Can this method be integrated with self-consistency decoding? For instance, could the majority voting result from multiple reasoning chains with the generated knowledge, lead to better outcomes?
4. The authors are encouraged to include more qualitative examples to compare the behavior between reasoning with a 0-shot prompt and reasoning with generated examples.

**Questions:**

The most important questions are mentioned in the "Weaknesses" section. Here is an additional question I am interested in, which may not necessarily be included in this paper.

Can a verification stage be added to filter out incorrect or irrelevant examples to further improve performance?

---

> ### Author Response · Authors · 2023-11-18
> **Thanks for your review, and our response**
>
> We sincerely thank the reviewer for insightful feedback. We have incorporated all suggestions in our paper. We thank the reviewer for describing our method as intuitive, well-motivated and demonstrating effectiveness across diverse datasets. We respond to the reviewer’s concerns and questions below.
>
> > Will the language model be easily distracted when the generated examples are irrelevant or incorrect?
>
> Great question. Below is the analysis of 50 correctly solved problems and 50 incorrectly solved problems from GSM8K+MATH. Overall, when the generated examples are irrelevant or incorrect, the LLM produces more wrong answers (22) than correct answers (15), but the gap is not large, indicating that LLM is not distracted to a critical extent.
> - 50 correctly solved problems:
>   - (6/50) Generated exemplars are irrelevant
>   - (9/50) Generated exemplars are relevant but incorrect
>   - (35/50) Generated exemplars are relevant and correct
> - 50 incorrectly solved problems:
>   - (10/50) Generated exemplars are irrelevant
>   - (12/50) Generated exemplars are relevant but incorrect
>   - (28/50) Generated exemplars are relevant and correct
>
> > Can current LLMs generate helpful examples for challenging questions?
>
> Yes, we observed that the LLM generates helpful exemplars for challenging problems. It can generate relevant, simpler problems and their solutions, which help the LLM solve more challenging target problems. For instance, the Codeforces task often presents challenging code problems, and the LLM generates relevant basic problems like typical prefix sum and dynamical programming problems, which help solving the challenging target problems. We have added examples to the appendix D.
>
> > Can this method be integrated with self-consistency decoding?
>
> This is a great suggestion. Yes, our method can be integrated with self-consistency and this indeed improves the performance:
> - GSM8K
>   - Ours: 77.8
>   - Ours + self-consistency: 85.3
> - MATH
>   - Ours: 37.3
>   - Ours + self-consistency: 46.0
>
> In particular, we find that self-consistency reduces the failure case of self-generating irrelevant examplers and getting distracted. This suggests that self-consistency and our analogical prompting can effectively complement each other.
>
> > The authors are encouraged to include more qualitative examples to compare the behavior between reasoning with a 0-shot prompt and reasoning with generated examples.
>
> Thank you for your suggestion. We have added examples comparing 0-shot and our method in the Appendix D.
>
>
> > Here is an additional question I am interested in, which may not necessarily be included in this paper. Can a verification stage be added to filter out incorrect or irrelevant examples to further improve performance?
>
> This is a very interesting idea. We experimented with this idea by sampling multiple exemplars, filtering out incorrect or irrelevant exemplars using GPT-4 as the verifier (i.e. prompting GPT-4), and then re-prompting the original LLM with the remaining exemplars to solve the target problem. We found this improves the task performance (e.g. 2% improvement in the MATH task).

---

> > ### Comment · Reviewer_ExxM · 2023-11-19
> > **Thanks for the response!**
> >
> > Thank you for the detailed experiments and explanations provided. However, after reviewing the rebuttal, my concerns regarding the efficacy of the generated examples have increased. As a result, **I have adjusted my score from 6 to 5**.
> >
> > I would like to request further clarification on the following points:
> >
> > 1. In my initial query, I asked for specific numbers regarding the sample distribution: how many samples are from the GSM8K dataset and how many are from the MATH dataset? This information is crucial for a comprehensive evaluation.
> >
> > 2. Regarding unsolved questions, it appears that Language Models (LLMs) can provide relevant and correct examples in 56% of cases. This leads to a critical question: why are LLMs unable to solve these questions? Does this suggest that it is inherently challenging for LLMs to learn from generated examples, indicating that they are not particularly effective at analogical reasoning? If this is the case, could you propose any directions or strategies for achieving further improvements in this area?

---

> ### Author Response · Authors · 2023-11-20
> **Thanks for your questions, and our clarification**
>
> Thanks for reading our response!
>
> > 1. how many samples are from the GSM8K dataset and how many are from the MATH dataset?
>
> It is 50%-50% ratio from GSM8K and MATH (paper Section 6.6).
>
> > 2. Regarding unsolved questions, it appears that Language Models (LLMs) can provide relevant and correct examples in 56% of cases. Why are LLMs unable to solve these questions?
>
> Below is the analysis of these unsolved problems where LLMs provided relevant and correct exemplars (full detail in paper Section 6.6):
>
> - (28/50) Generated exemplars are relevant and correct, but LLM fails to solve the new problem:
>   - (12/50) A generalization gap between the generated exemplars and the new problem.
>   - (8/50) Overreliance on specific exemplars, such as copying.
>   - (8/50) Other issues, such as calculation errors.
>
> Remaining 22/50 unsolved problems:
> - (10/50) Generated exemplars are irrelevant
> - (12/50) Generated exemplars are relevant but incorrect
>
> As this suggests, among the unsolved problems, the error modes are spread evenly (irrelevant exemplars, incorrect exemplars, correct exemplars with generalization gap, other LLM mistakes). Therefore, we do not think there is a particular weakness such as "it is inherently challenging for LLMs to learn from generated examples".
>
> Regarding future directions to achieve further improvements:
>
> - To generate more relevant and correct exemplars: as the reviewer suggested, we can add a verification stage to filter out incorrect or irrelevant examples
> - To generate exemplars with less generalization gap: we can include additional instruction in the prompts so that we generate exemplars sequentially from simpler exemplars to harder exemplars that are closer to the target problem.
>
> **Most importantly**, please note that this analysis is an **error analysis**, which **only looks at unsolved problems** (only a part of the total event space) to see "among unsolved problems, how often the LLMs provided relevant and correct exemplars". In the majority of situations, the LLMs provided relevant and correct exemplars, and solved the target problem correctly, i.e., performed analogical reasoning successfully.
>
> We hope that this response resolves the confusion or concerns the reviewer may have. Please feel free to let us know if you have any further questions!

---

> > ### Comment · Reviewer_ExxM · 2023-11-20
> > **More Discussions**
> >
> > 1. I'm interested in separate analyses for GSM8K and MATH.
> >
> > 2. Regarding the table:
> > 50 correctly solved problems:
> > (6/50) Generated exemplars are irrelevant
> > (9/50) Generated exemplars are relevant but incorrect
> > (35/50) Generated exemplars are relevant and correct
> > 50 incorrectly solved problems:
> > (10/50) Generated exemplars are irrelevant
> > (12/50) Generated exemplars are relevant but incorrect
> > (28/50) Generated exemplars are relevant and correct
> >
> > This indicates exemplars are useful in 56% of unsolved problems and 70% of solved ones. However, in 30% of solved problems, LLMs disregard the exemplars. Additionally, LLMs fail to benefit from exemplars in at least 56% of unsolved problems. This suggests difficulty in altering LLMs' reasoning from their trained biases. Thus, I am not confident that the current LLMs are already good analogical reasoners.
> >
> > In the end, I think it is an interesting direction. But current experiments cannot fully convince me of its efficacy. Thus, I will keep the score **5**.

---

> ### Author Response · Authors · 2023-11-20
> **Thanks for your questions, and our clarification**
>
> These are great questions.
>
> > I'm interested in separate analyses for GSM8K and MATH.
>
> GSM8K:
> - 25 solved problems
>   - (2/25) Generated exemplars are irrelevant
>   - (4/25) Generated exemplars are relevant but incorrect
>   - (19/25) Generated exemplars are relevant and correct
> - 25 unsolved problems
>   - (4/25) Generated exemplars are irrelevant
>   - (5/25) Generated exemplars are relevant but incorrect
>   - (16/25) Generated exemplars are relevant and correct
>     - (7/25) A generalization gap between the generated exemplars and the new problem (e.g., harder).
>     - (5/25) Overreliance on specific exemplars, such as copying.
>     - (4/25) Other issues, such as calculation errors.
>
> MATH:
> - 25 solved problems
>   - (4/25) Generated exemplars are irrelevant
>   - (5/25) Generated exemplars are relevant but incorrect
>   - (16/25) Generated exemplars are relevant and correct
> - 25 unsolved problems
>   - (6/25) Generated exemplars are irrelevant
>   - (7/25) Generated exemplars are relevant but incorrect
>   - (12/25) Generated exemplars are relevant and correct
>     - (5/25) A generalization gap between the generated exemplars and the new problem (e.g., harder).
>     - (3/25) Overreliance on specific exemplars, such as copying.
>     - (4/25) Other issues, such as calculation errors.
>
> > in 30% of solved problems, LLMs disregard the exemplars
>
> This means that LLM solved the target problems correctly even when the generated exemplars were irrelevant or incorrect. This is reasonable – it is known that LLMs can few-shot learn even from wrong in-context exemplars (https://arxiv.org/abs/2202.12837).
>
> >  LLMs fail to benefit from correct exemplars in at least 56% of unsolved problems. current experiments cannot fully convince me of its efficacy
>
> We would like to clarify that this is reasonable. Our work does **not** suggest that when the exemplars are correct, LLMs should always produce the correct answer. For example, few-shot CoT always uses correct exemplars, but can fail to solve the target problem too – in fact, fails more often than our method (Table 1). **Our main contribution is that we show our method outperforms manual few-shot CoT**, i.e., LLMs benefit more from our self-generated exemplars than hand-written exemplars.

---

> > ### Comment · Reviewer_ExxM · 2023-11-22
> > **More comments**
> >
> > If the paper aims to demonstrate better automatic prompt engineering, then you need to provide more comparisons with existing works, such as "Large Language Models Are Human-Level Prompt Engineers" and its follow-up. Or you need to include experiments to demonstrate when LLMs are analogical reasoners and when not. **At this time, the paper is not strong enough to get in.**

---

> > > ### Author Response · Authors · 2023-11-23
> > > **Author response to comments**
> > >
> > > Thank you for your comment.
> > >
> > > > If the paper aims to demonstrate better automatic prompt engineering
> > >
> > > We would like to clarify that our setting is fundamentally different from existing works on automatic prompt engineering. First, automatic prompt engineering works (e.g., "Large Language Models Are Human-Level Prompt Engineers" mentioned in the review) typically require access to a training set to compute the score of each prompt. In contrast, our analogical prompting method does not require training data. Additionally, automatic prompt engineering works focus on modifying prompt phrases (e.g., instructions) to increase the task accuracy, whereas our analogical prompting calls the LLM to self-generate exemplars to be used in chain-of-thought reasoning. We hope this clarification addresses your concerns.

---

### Official Review · Reviewer_DVcm · 2023-11-01

**Soundness:** 2 fair
**Presentation:** 3 good
**Contribution:** 2 fair
**Rating:** 5
**Confidence:** 3

**Summary:**

The authors proposes a new prompting paradigm, which has three phases:
1. Related knowledge retrieval
2. Exemplar generation
3. Answer prompting.

With this paradigm, the prompting on math and reasoning required tasks has an average accuracy gain of +4%.

**Strengths:**

Originality: 3/5

This paper aims to solve the problem that the few-shot prompting schema requires manually collected examples via a template-based method.
Although there are quite a few previous works on prompt templates and knowledge retrieval, such as recitation-augmented models, this work focuses more on reasoning-based problem-solving.

Quality: 2.5/5

This work has performed studies on quite a few benchmarks, including GSM8K and MATH, code generation in Codeforces, and other reasoning tasks in BIG-Bench. However, the experiment setup is a bit weird, as some of the studies include the in-context demonstration of generating examples, some do not; some adopt with knowledge paradigm, and some do not. A minor concern is that this study does not include the GPT-4 performance.

Clarity: 3.5/5

Overall the paper is well written and easy to follow. The examples are quite illustrative but may be a bit repetitive, as Figure 3 seems to already include all the information that Figure 2 contains.

Significance: 3/5

This method proposes a new prompting schema for leveraging the language model as a knowledge base. However, this exemplar generation procedure does not provide an in-depth guarantee or study on the quality of generated examples.

**Weaknesses:**

1. The experiment setup could be better.

**Questions:**

1. It would be nice to conceptually compare the work against the neural symbolic method [1].

[1] Zhang, Hanlin, et al. "Improved logical reasoning of language models via differentiable symbolic programming." arXiv preprint arXiv:2305.03742 (2023).

---

> ### Author Response · Authors · 2023-11-18
> **Thanks for your review, and our response**
>
> We sincerely thank the reviewer for constructive feedback. We have incorporated all suggestions in our paper. We appreciate that the reviewer describes our work as offering a novel and effective prompting strategy for LLMs. We respond to the reviewer’s concerns and questions below.
>
> > experiment setup is a bit weird, as some of the studies include the in-context demonstration of generating examples and knowledge paradigm. A minor concern is that this study does not include the GPT-4 performance.
>
> We included the specific setups, such as in-context demonstration and knowledge generation, for the reasons outlined below. We appreciate the reviewer's suggestion on improving our presentation, and we've addressed it in the updated paper.
> - Re: In-context demonstration. We included this setup to experiment with our approach across various LLMs. GPT3.5-turbo and GPT-4 were trained with extensive instruction tuning / RLHF, and they do not need any in-context demonstration to be able to generate exemplars and solutions. On the other hand, Davinci models were trained with less instruction tuning / RLHF, and in such cases, in-context demonstration boosts their performance (Table 1).
> - Re: Self-generation of knowledge. This technique supplements our primary approach of self-generating exemplars. This is especially useful for complex tasks like code generation, where high-level knowledge complements low-level exemplars. We also evaluated this knowledge generation technique in other tasks like GSM8K and obtained performance boosts (0.5%).
> - Re: GPT-4. We reported GPT-4 performance mainly for the challenging Codeforces 2023 task (Table 2), considering that in other tasks like GSM8K, the base GPT-4 performance was already high (e.g. 92% accuracy), possibly because these task datasets were seen during GPT-4 training. Nevertheless, in both cases, our method provides performance boosts over the baseline CoT for GPT-4: 4% in Codeforces and 1% in GSM8K.
>
> In our updated paper, we present all models (including GPT-4) and setups (exemplar generation and knowledge generation) in a cohesive manner.
>
> > does not provide study on the quality of generated examples.
>
> Below is the manual analysis of the quality of generated examples, based on 50 correctly solved problems and 50 incorrectly solved problems from GSM8K+MATH:
> - 50 correctly solved problems:
>   - (6/50) Generated exemplars are irrelevant
>   - (9/50) Generated exemplars are relevant but incorrect
>   - (35/50) Generated exemplars are relevant and correct
> - 50 incorrectly solved problems:
>   - (10/50) Generated exemplars are irrelevant
>   - (12/50) Generated exemplars are relevant but incorrect
>   - (28/50) Generated exemplars are relevant and correct
>
> Overall, we find that the majority of the generated exemplars relevant and correct. We have added the detailed analysis to our paper.
>
>
> > It would be nice to conceptually compare the work against the neural symbolic method [1].
>
> Thank you for pointing us to the related work. We have added a citation to the neural symbolic work in our paper. Our approach, analogical prompting, is distinct yet complementary to neural symbolic methods. We present a high-level LLM reasoning strategy, demonstrating how generating related exemplars through analogy aids in problem-solving. Neural symbolic methods could further complement the problem-solving stage, which is an interesting avenue for future research.

---

### Author Response · Authors · 2023-11-18
**General Response to Reviewers**

We thank all the reviewers for their insightful and constructive feedback. We appreciate the reviewers’ positive comments that our prompting method is novel (ey56), well-motivated (ExxM), and effective across diverse tasks (DVcm) and that our work provides valuable design principles and ablation studies (fGoM).

We have addressed each reviewer’s concerns and questions in the individual responses below. We also updated our paper to address requests such as analysis on the quality of generated exemplars, more examples of prompts and model predictions, self-consistency results, additional ablations and references.

Please do not hesitate to contact us for any further questions or clarifications. Thank you for your time and consideration.

---

### Author Response · Authors · 2023-11-22
**Author Response to Reviewer Concerns**

Several reviewers appear to have a misunderstanding of our work, and we would like to respectfully provide clarification here.

Our analogical prompting method works as follows: given a problem, our method self-generates relevant exemplars and then solves the target problem. The evaluation demonstrates that without manually labeling the exemplars, our analogical prompting method outperforms both zero-shot prompting and few-shot chain-of-thought prompting with hand-written or retrieved exemplars.

The reviewers questioned our work because: (a) the LLM sometimes fails to solve the target problem even if the self-generated exemplars are correct; and (b) the LLM sometimes correctly solves the target problem even if the self-generated exemplars are incorrect. We would like to clarify that **these observations hold for all kinds of prompting methods, and do not undermine the efficacy of our method**. The main contribution of our work is to show that **LLMs achieved better performance with their self-generated exemplars than hand-written exemplars. The fact that they do not achieve 100% accuracy when given correct exemplars does not invalidate our approach.**

> Case (a) "the LLM sometimes fails to solve the target problem even if the self-generated exemplars are correct"

The standard few-shot CoT always uses correct exemplars, but can fail to solve the target problem too – in fact, it fails more often than our method (Table 1).

> Case (b) "the LLM sometimes correctly solves the target problem even if the self-generated exemplars are incorrect"

This observation is also not unique to our method, and prior works have also shown that LLMs typically achieve positive task accuracies even when given wrong in-context exemplars (e.g., https://arxiv.org/abs/2202.12837, https://arxiv.org/abs/2209.07686, etc.).

Thank you once again for your time and consideration.

---

### Meta-Review · Area_Chair_K74g · 2023-12-13

**Metareview:**

**Paper Summary:**

This paper introduces "Analogical Prompting," a new prompting method for large language models (LLMs). This approach prompts LLMs to first generate relevant examples or knowledge, and then generate solutions. The method was evaluated on mathematical problem-solving (GSM8K, MATH), code generation (Codeforces), and other reasoning tasks (BIG-Bench). The results demonstrated improvements over previous prompting techniques.

**Strengths:**

1. Simplicity and Intuitiveness: The method is simple and intuitive (ExxM, fGoM, ey56).
2. Good Results: The proposed method shows surprising improvements across an array of datasets compared to traditional few-shot prompting, which relies on training examples (ExxM, fGoM).


**Weaknesses:**

1. Limited Technical Contribution: The method primarily contributes to prompt engineering and its effectiveness might diminish as LLMs get better over time (fGoM).
2. Concerns Over Generated Examples: There are raised questions about the potential inaccuracies in self-generated examples and their ability to distract or mislead the model. Interestingly, correct solutions are sometimes derived from incorrect examples, posing questions about whether the LLMs is truly doing analogical reasoning (ExxM, fGoM).

**Decision:**

Despite the reviewers' concerns regarding the approach being an instance of prompt engineering, this method is simple yet seems to bring gains in LLMs, and I think it has the potential to be used as baselines in future works. Therefore, I recommend accepting this paper as a poster. However, I urge the authors to address the reviewers' feedback in the next iteration of their work, specifically focusing on the inclusion of more qualitative examples and error analyses. In addition, it is surprising that analogical reasoning can outperform few-shot CoT, considering it doesn't rely on training examples, unlike few-shot CoT. It would be nice to discuss the intuitions behind why it works.

**Justification For Why Not Higher Score:**

Based on reviews, the proposed approach has limited technical contributions and only brings small gains on some datasets.

**Justification For Why Not Lower Score:**

This method is simple and intuitive, and has the potential to be used by future works as baselines.

---

### Decision · Program_Chairs · 2024-01-16

Accept (poster)